# Evaluation of primary HPV-based cervical screening among older women: Long-term follow-up of a randomized healthcare policy trial in Sweden

**Qingyun Yao** [1] *, **Jiangrong Wang** [1], **K. Miriam Elfström** [1], **Björn Strander** [2], **Joakim Dillner** [1,3], **Karin Sundström** [1,3]

**1** Center for Cervical Cancer Elimination, Department of Clinical Science, Intervention and Technology, Karolinska Institutet, Huddinge, Sweden, **2** Institute of Clinical Sciences, Sahlgrenska Academy, University of Gothenburg, Gothenburg, Sweden, **3** Medical Diagnostics Karolinska, Stockholm, Sweden

* qingyun.yao@ki.se

## Abstract

### Background

Evidence on invasive cervical cancer prevention among older women is limited, especially with the introduction of human papillomavirus (HPV)-based screening and longer interval. We conducted a long-term follow-up of the first phase of a randomized healthcare policy trial in cervical screening, targeting women aged 56 to 61 years old, to investigate the effectiveness of primary HPV-based screening in preventing invasive cervical cancer (ICC) and the safety of extending screening interval.

### Methods and findings

The randomized healthcare policy trial of primary HPV-based cervical screening targeted women residing in Stockholm-Gotland region during 2012 to 2016, aged 30 to 64 years. The trial aimed to investigate the detection rate of cervical intraepithelial neoplasia grade 2 or worse (CIN2+) within 24 months and long-term protection against invasive cervical cancer, comparing primary HPV-based screening to primary cytology-based screening. The initial phase of the trial, which was the focus of this study, targeted women aged 56 to 61 years old in 2012 to 2014 who were randomized to primary cytology arm ($n = 7,401$) or primary HPV arm ($n = 7,318$). We used national registries to identify the subsequent cervical tests and all histopathological diagnoses including ICC before December 31, 2022. We calculated cumulative incidence, incidence rate (IR) and IR ratio (IRR) of ICC, by baseline test result. Furthermore, we calculated longitudinal sensitivity and specificity for detecting cervical intraepithelial neoplasia grade 2 or worse (CIN2+) by receipt of primary cytology or primary HPV test for the recommended screening intervals in this age group. We found that the IR of ICC among women in the primary HPV arm was 7.2/100,000 person-years (py) and 3.0 for women who tested HPV negative, compared to 18.4/100,000 py among women in the primary cytology arm and 18.8 for women who tested cytology negative. We further found that

**Data Availability Statement:** Our data contains potentially identifying patient information, thus

according to Swedish law the authors are not able to make the dataset publicly available. The data is deposited with the Swedish Cervical Screening Registry (www.nkcx.se) and can be requested at info@nkcx.se (administrator Sara Nordqvist Kleppe). One can vist http://www.nkcx.se/research_e.htm for detailed information about how to apply for access. The code used in the analysis is available from Github [https://github.com/qingyunyao/Code-for-fas1-follow-up/releases/tag/v1.0.0] and archived in Zenodo [https://zenodo.org/doi/10.5281/zenodo.13285310].

**Funding:** This study is funded by the Swedish Cancer Society (Grant number: 21 1690 Pj, https://www.cancerfonden.se/forskning) and the Cancer Research Funds of Radiumhemmet (Grant number: 234172, https://rahfo.se/for-forskare/forskningsanslag/), received by KS, and China Scholarship Council, the latter received by QY (File no. 202206160010, https://www.csc.edu.cn). The funders had no role in study design, data collection and analysis, decision to publish, or preparation of the manuscript.

**Competing interests:** The authors have declared that no competing interests exist.

**Abbreviations:** AC, adenocarcinoma; ASC, adenosquamous carcinoma; ASCUS, atypical squamous cells of undetermined significance; CI, confidence interval; CIN2+, cervical intraepithelial neoplasia grade 2 or worse; HPV, human papillomavirus; ICC, invasive cervical cancer; IR, incidence rate; IRR, incidence rate ratio; LSIL, low-grade squamous intraepithelial lesion; SCC, squamous cell carcinoma.

the overall point estimate for the risk of ICC over 10 years of follow-up among women in the primary HPV arm was 0.39 compared to women in the primary cytology arm, but this was not statistically significant (IRR: 0.39; 95% confidence interval, CI [0.14, 1.09]; $p = 0.0726$). However, among women with a negative test result at baseline, women in the primary HPV arm had an 84% lower risk of ICC compared to women in the primary cytology arm (IRR: 0.16; 95% CI [0.04, 0.72]; $p = 0.0163$). Moreover, primary HPV testing had a higher sensitivity for detecting CIN2+ within a 7-year interval than primary cytology testing within a 5-year interval (89.6% versus 50.9%, $p < 0.0001$). We were limited by a partial imbalance of invitations during the follow-up between the 2 arms which may have led to an underestimation of the effectiveness of primary HPV-based screening.

## Conclusions

In this study, we observed that women over 55 years of age who received a primary negative HPV test result had substantially lower risk of CIN2+, and ICC, compared to women who received a primary negative cytology result. This should apply even if the screening interval were prolonged to 7 years.

## Trial Registration

NCT01511328.

## Author summary

### Why was this study done?

- Cervical screening in women above the age of 55 to 60 years is challenging, and invasive cancer still occurs in this age group.

- Controversies remain on the choice of optimal test, the interval, the upper age limit, and criteria for discontinuing screening.

### What did the researchers do and find?

- We performed a long-term follow-up of a randomized healthcare policy trial including 14,719 women in Stockholm-Gotland region of Sweden, comparing the effectiveness of primary HPV-based screening and primary cytology-based screening among women aged 56 to 61 years.

- Women who tested negative by primary cytology at the age of 56 to 61 still had a noticeable risk of cervical cancer over the study period, whereas women who tested negative by primary HPV had an 84% lower risk of cancer.

- Primary HPV testing had higher longitudinal sensitivity in detecting precancerous lesion at 5 years, 7 years, and even at 10 years, respectively, after the baseline test compared to primary cytology testing.

**What do these findings mean?**

- For women older than 55 to 60, those who receive a negative HPV result are at substantially lower risk of cervical cancer compared to those who receive a negative cytology result.

- The screening interval could conceivably be extended to 7 to 10 years, when an HPV-based cervical screening policy is adopted.

- We conclude that for women to exit cervical screening at the age of around 60 with only a primary cytology negative result is not optimal.

- We may have slightly underestimated the overall performance of primary HPV-based screening due to women in cytology arm ultimately having undergone a more intensive screening.

## Introduction

Cervical cancer incidence displays a bimodal pattern in a screened population, with a first peak around women of 35 to 39 years old (incidence rate, IR = 19.2/100,000 person-years) and a second peak among women aged 65 to 69 years old (IR = 16.5/100,000 person-years) [1,2]. To achieve the goal of eliminating cervical cancer as a public health problem, defined as an incidence below 4/100,000 person-years [3], it is of great importance to optimize screening also among older women.

The relatively higher incidence of cervical cancer and cervical intraepithelial neoplasia grade 2 or worse (CIN2+) among older women has been associated with suboptimal screening participation, diagnostic difficulties due to morphological changes due to mucosal atrophy due to menopause, and controversy in screening upper age limit [1,4]. An efficient cervical cancer screening program can provide early detection of precancerous lesions which can lead to early treatment intervention and eventually reduction in risk of invasive cervical cancer (ICC), as well as providing early detection of manifest ICC. Regardless of the known low sensitivity of cytology-based cervical screening among older women [5], however, only 35% of countries worldwide currently endorse human papillomavirus (HPV)-based screening in their national recommendations [6]. Only 5 European countries had started implementing primary HPV-based screening by 2019 [7]. There is also a lack of evidence of when it is appropriate to exit the screening. Most countries recommend women exit screening around 60 to 70 years old [6]. The recommended criterion of exiting the program differs between countries. Based on recommendations from American Cancer Society, only women without a history of CIN2+ within past 25 years and with adequate negative screening history within the past 10-year period can discontinue screening [8]. The European guidelines suggest women with 1 negative HPV test when arriving at the upper age limit (60 or 65 years) can discontinue screening [9]. Current screening strategy recommendations for women over 50 years are mainly based on the evidence from cytology testing, using precancer as the outcome, and women of younger age [10]. There are few studies investigating long-term follow-up of primary HPV-based screening in women aged over 50 years, however, as these studies are either without age stratification in their results, or only using precancer lesions as the main outcome [11–13]. Thus,

trials with ICC as outcome to evaluate screening strategies and safety among older women are needed.

In 2012, we began implementing a 2-phase randomized healthcare policy trial inviting all eligible women aged 30 to 64 years residing in the capital Stockholm-Gotland region. The first phase, reported here, targeted women aged 56 to 61 years during 2012 to 2014 and the second phase targeted women aged above 30 years. The goal of both phases, respectively, was to investigate the detection rate of CIN2+ within 24 months and long-term protection against ICC, comparing primary HPV-based screening to primary cytology-based screening. At the time of the trial initiation, the Swedish guidelines for cervical screening recommended to screen women aged over 50 years with primary cytology test and a 5-year interval. The upper limit age for screening at that time was set to 60 years old, which resulted in women receiving the exit testing at the age around 56 to 61 years. Based on this, the first phase of the randomized healthcare policy trial aimed to compare the effectiveness of primary HPV testing and primary cytology testing as exiting test in this age group [14,15]. We have previously reported similar baseline detection rates of CIN2+ whether using primary HPV testing or primary cytology among older women in a non-inferiority randomized trial of exit testing with HPV compared to exit testing with cytology, in 2012 in Sweden [14]. Several studies have subsequently confirmed these results [16,17].

In 2017, updated guidelines recommending primary HPV-based screening were implemented in Stockholm, where women aged over 50 years old started receiving primary HPV-based screening with a 7-year interval and with the upper age limit set at 64 years. The guidelines were also clarified to state that all women should receive HPV testing rather than cytology at exit from screening. This yields a unique opportunity to study in an ethically acceptable manner the long-term incidence of cervical cancer among older women who received HPV versus cytology, as all women in the original cytology arm have since already been offered HPV as part of a subsequent program effort. To investigate effectiveness associated with the change of primary test method and prolonged time interval, we thus here present a long-term follow-up of the randomized healthcare policy trial implemented in 2012, comparing the 10-year effectiveness of primary HPV-based screening and primary cytology-based screening, respectively, in detecting CIN2+ and preventing ICC among women aged 56 to 61 years old in the capital region of Sweden. We also compared the effectiveness of primary HPV and primary cytology-based screening among women with or without previous abnormal screening history, respectively.

## Material and methods

### Study population and study procedure

We performed a long-term follow-up of a randomized healthcare policy trial [18] established in 2012 in Stockholm-Gotland region. During January 2012 to May 2014, women who were aged 56 to 61 years (age defined by birth year, women were born between January 1, 1951 and December 31, 1958) and resident in the Stockholm-Gotland region of Sweden were invited to their last screening test based on the cervical cancer screening policy before 2017 [15]. Participating women and midwives who took the samples were blinded for the sample analysis method. After receiving the samples, we randomized the samples in the lab into 2 groups based on the last digit of Swedish personal identity number of the participant: (1) Cytology arm: primary cytological test and with triage HPV analysis for women with low-grade cytological abnormality (atypical squamous cells of undetermined significance, ASCUS, or CIN1/low-grade squamous intraepithelial lesion, LSIL), according to the routine program; and (2) HPV-arm: primary HPV test and triage cytological analysis for women with positive HPV results.

The detailed screening protocol of this trial was reported in our previous study [14]. Detailed information on the diagnostic algorithm can also be found in the Supporting information (S1 Fig and Screening Protocol in S1 Text). The protocol of this randomized healthcare policy trial was discussed by a committee of specialized gynecologists and in a national hearing before it was implemented. The trial was approved by the Swedish Ethical Review Authority (DNR 2011/1298-31/3) and registered at www.clinicaltrials.gov (registration number NCT01511328). The ethical review board decided participating in screening following an invitation constituted appropriate consent for participation such that no further consent was required. Further details on the study design and follow-up of test-positive women have been described previously [14,19].

We used the Swedish National Cervical Screening Registry (NKCx), which has a full coverage nationally of cervical cancer screening results since 1995 [20], including all organized screening program tests, all non-organized/opportunistic tests, all indicated tests due to, e.g., clinical symptoms, and all cervical histopathology tests and results, to retrieve all the screening tests and test results from the trial and subsequently during our follow-up. We identified 14,719 women who participated in the trial (cytology arm, $n = 7,401$; HPV arm, $n = 7,318$). We excluded women who had hysterectomy registered in the register or opted out of screening before the randomized trial ($n = 3$) and women with invalid test result at baseline in the trial ($n = 3$, all in the HPV arm). Follow-up started at the first sample taken between January 1, 2012 and May 31, 2014. We categorized the baseline test results as HPV positive (regardless of HPV type), HPV negative, cytology positive (ASCUS+), and cytology negative. We identified CIN2+ (includes CIN2, CIN3, Adenocarcinoma in situ, ICC) diagnosed by a histopathology test in the NKCx. In the analysis of histopathology-confirmed CIN2+ as outcome, the follow-up time was defined as the time elapsed from the baseline test to either the first histopathological diagnosis of CIN2+, or the date of the last registered test, including cytology, HPV, and histopathology before December 31, 2022. In the analysis of ICC as outcome, we used the Swedish National Quality Register for Gynecological Cancers (QGCR), which has the detailed information on cervical cancer diagnosis nationwide since 2011 [21], to identify ICC cases, including squamous cell carcinoma (SCC), adenocarcinoma (AC), adeno-squamous carcinoma (ASC), and other more rare histological types still deemed as HPV-associated [22]. The register uses International Classification of Diseases-10th Revision (ICD-10) code C53 to identify ICC. End of ICC follow-up was defined as the first cancer diagnosis in the QGCR, the date of emigration, death, or total hysterectomy in the NKCx, or December 31, 2022, whichever came first.

## Statistical analysis

Our primary outcome was the incidence of ICC and our secondary outcome was the incidence of histopathology-confirmed CIN2+ lesions. We used Kaplan–Meier curves to calculate the survival probabilities and generated the standard error of survival probability by the Greenwood formula. We further used 1-the Kaplan–Meier curve to calculate the cumulative incidence of histopathology-confirmed CIN2+ and ICC, by baseline test results. We performed log-rank tests comparing the survival curves of histopathology-confirmed CIN2+ and ICC among women with different baseline result. We calculated follow-up participation within 3 year, 6 years, and 10 years, respectively. The participation of follow-up was defined as any new test registered in NKCx after the baseline screening test. We calculated the number of first organized screening test of participants after the baseline test by calendar year. We calculated the longitudinal characteristics (sensitivity and specificity) of primary HPV and primary cytology testing to identify histopathology-confirmed CIN2+ at 3, 5, 7, and 10 years with

conditional weighting, as the censoring of cases is test result dependent [23]. The longitudinal characteristics are important indicators in evaluating screening strategies and show the ability of the primary test method to detect or predict the presence or development of precancer [24]. We chose the time points based on former and current screening strategies (3-year interval for women under 50 years, 5-year interval before 2017, 7-year interval after 2017 for women over 50 years, and 10-year interval as used in some settings internationally). We used a two-sample test of proportions assuming a binomial distribution to compare the test sensitivity of primary HPV testing at 5-year, 7-year, and 10-year intervals to primary cytology testing sensitivity at a 5-year interval.

We calculated the incidence rate (IR) and 95% confidence intervals (CIs) by baseline test results. To compare the effectiveness of the primary test methods, we also calculated the incidence rate ratio (IRR) and 95% CI of HPV arm compared to cytology arm. We found an imbalance of follow-up participation between the 2 study arms. We calculated the number of women who had first organized screening test after baseline test by calendar year in each arm. Women in cytology arm who had sample taken in 2012 or 2013 started to receive second round of organized screening after 3 years and women in HPV arm was after 5 years (Table A in S1 Text). Therefore, we used Poisson regression adjusting for sample year to explore the adjusted IRR of histopathology-confirmed CIN2+ and ICC between primary test methods.

After a national policy change in 2017, it was clarified that all women until age 64 years should undergo primary HPV-based cervical screening instead of cytology. Thus, all women in Region Stockholm-Gotland within the new screening age range were re-invited into the reformed screening program in a prevention effort free-standing from our original trial. Around 85% (6,166/7,256) of women tested negative in cytology arm and 78% (5,423/6,909) in HPV arm thus received at least one more organized screening test after the here defined trial baseline test (Table A in S1 Text). As a result, we were not able to test the effectiveness of HPV and cytology testing as the exiting test at age of 56 to 61 years, but we were able to perform a subgroup analysis with women who tested negative at baseline. We calculated the proportion of turning HPV positive, the incidence rate of histopathology-confirmed CIN2+, and the incidence rate of ICC among women with organized screening test or without any organized screening test (including women with no test or opportunistic/indicated test) after baseline test. To investigate the effectiveness of primary HPV-based screening compared to primary cytology-based screening among high-risk women, we performed subgroup analyses among women with or without previous abnormality; detailed information is provided in Supporting information (Supplementary Statistical Analysis in S1 Text).

To calculate the incidence of the first occurrence of histopathology-confirmed CIN2+, all analyses using histopathology-confirmed CIN2+ as an outcome excluded women with histopathological diagnosed CIN2+ before the recruitment ($n = 294$) except in the subgroup analysis. Analyses were performed using SAS 9.4 and STATA 18. This study is reported as per the Consolidated Standards of Reporting Trials (CONSORT) guideline (S1 CONSORT Checklist). More detailed information on study implementation and analysis is provided in the study protocol (S1 Protocol) and analysis plan (S1 Statistical Analysis Plan).

## Results

Among the 14,713 women included in the final analyses, there were 6,909 with primary HPV negative result, 404 with primary HPV positive result, 7,256 with primary cytology negative result, and 144 with primary cytology positive result (Fig 1). The average age of women at the time of recruitment was 58.1 years old. The median follow-up time for CIN2+ was 6.1 years in cytology arm and 5.0 years in HPV arm. For invasive cervical cancer, it was 9.8 years in both

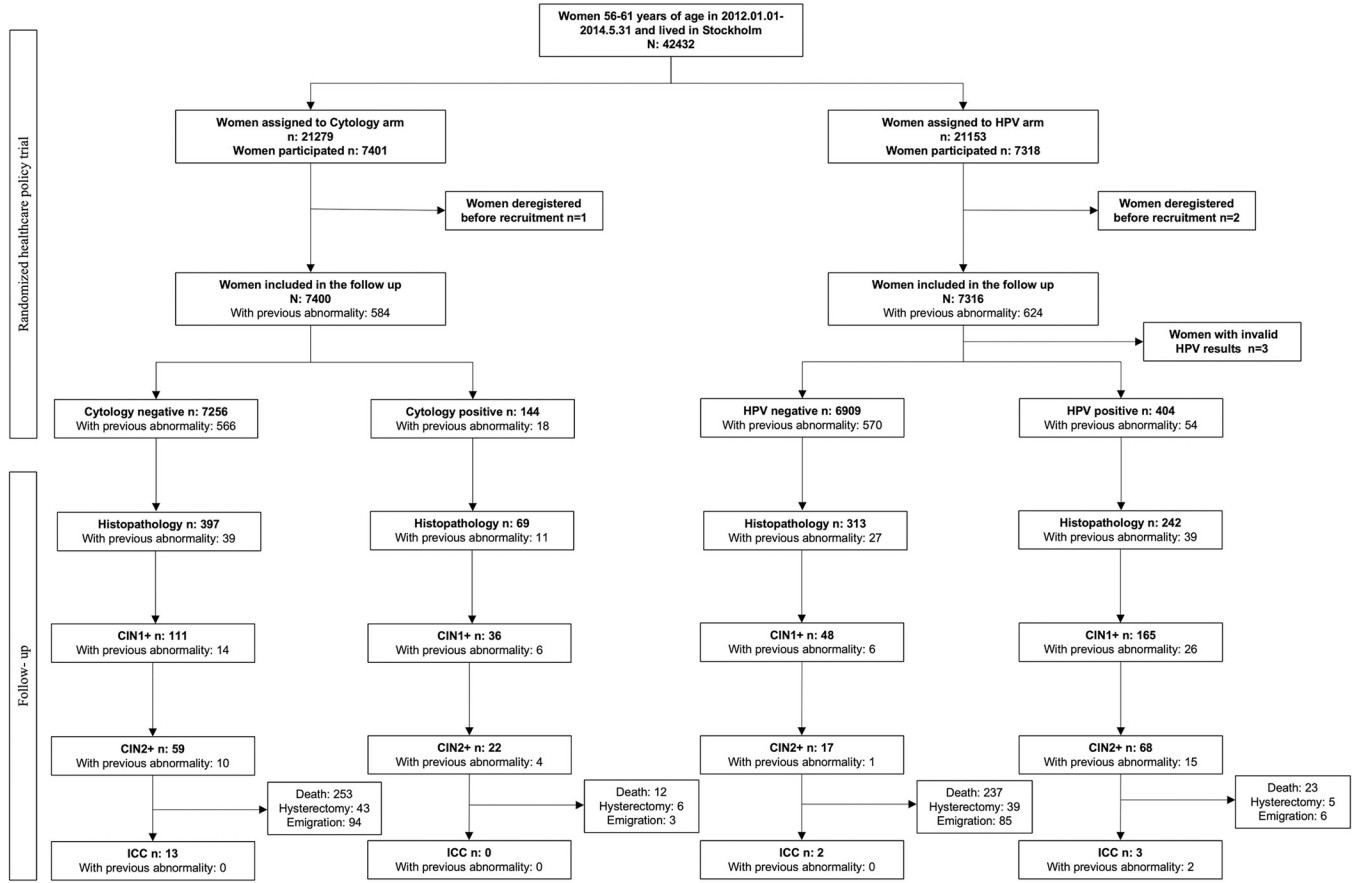

**Fig 1. Flow chart of the study population.** HPV, human papillomavirus; CIN1+, cervical intraepithelial neoplasia grade 1 or worse; CIN2+, cervical intraepithelial neoplasia grade 2 or worse; ICC, invasive cervical cancer; all the lesions were diagnosed by histo-pathological test; end of follow-up: 2022-12-31.

arms. The median number of screening tests was 2 for the whole cohort. The cytology arm had a median of 1 cytology test and 1 HPV test, while the HPV arm had 0 cytology tests and 2 HPV tests. Only 8.2% (1,208/14,713) of the whole cohort had a previous cervical abnormality (Table 1).

There were 466 out of 7,400 (6.3%) women in cytology arm and 555 out of 7,313 (7.6%) women in HPV arm who received at least 1 histopathology test (Fig 1). We identified 17, 68, 59, and 22 cases of histopathology-confirmed CIN2+ among primary HPV negative, primary HPV positive, primary cytology negative, and primary cytology positive women during the follow-up, respectively. We identified 2 ICC cases in primary HPV negative women, 3 in primary HPV positive women, 13 in primary cytology negative women, and no ICC cases in primary cytology positive women during the follow-up.

The cumulative incidence rates of CIN2+ and ICC are presented in Fig 2. No differences were found between 2 arms. The cumulative incidences of CIN2+ and ICC were all significantly higher among baseline primary cytology negative women compared to baseline primary HPV negative women (Fig 3, all $p < 0.05$). No difference was detected among positive groups (Fig 3).

The sensitivity of the primary HPV test was 98.1% at 3 years, 95.7% at 5 years, 89.6% at 7 years, and 82.1% at 10 years for the outcome of CIN2+. The sensitivity of primary cytology test at 5 years was only 50.9%. The sensitivity of primary HPV test detecting CIN2+ across years

**Table 1. Characteristics of the screening population, in total, by screening method, and by baseline results.**

| | All population | Cytology arm | HPV arm | Cytology Negative | Cytology positive | HPV negative | HPV positive |
|---|---|---|---|---|---|---|---|
| N | 14,713 | 7,400 | 7,313 | 7,256 | 144 | 6,909 | 404 |
| Mean age in years (SD) | 58.1 (1.31) | 58.1 (1.3) | 58.1 (1.3) | 58.1 (1.3) | 58.0 (1.3) | 58.1 (1.3) | 58.1 (1.4) |
| Median CIN2+ follow up (Q1–Q3) (years) | 5.2 (4.7–8.5) | 6.1 (3.7–8.7) | 5.0 (4.8–7.9) | 6.1 (3.7–8.7) | 5.9 (1.5–8.4) | 5.0 (4.8–7.8) | 6.0 (3.1–8.4) |
| Median cancer follow up (Q1–Q3) (years) | 9.8 (8.9–10.3) | 9.8 (8.9–10.3) | 9.8 (8.9–10.3) | 9.8 (8.9–10.3) | 9.3 (8.8–10.3) | 9.8 (8.9–10.3) | 9.8 (9.0–10.3) |
| Sample year | | | | | | | |
| 2012 | 5,683 (38.6%) | 2,817 (38.1%) | 2,866 (39.2%) | 2,761 (38.0%) | 56 (38.9%) | 2,710 (39.2%) | 156 (38.6%) |
| 2013 | 5,733 (30.9%) | 2,908 (39.3%) | 2,825 (38.6%) | 2,857 (39.4%) | 51 (35.4%) | 2,652 (38.4%) | 173 (42.8%) |
| 2014 | 3,297 (22.4%) | 1,675 (22.6%) | 1,622 (22.2%) | 1,638 (22.6%) | 37 (25.7%) | 1,547 (22.4%) | 75 (18.6%) |
| Median number of tests (range) | 2 (1–18) | 2 (1–18) | 2 (1–17) | 2 (1–12) | 4 (1–18) | 2 (1–11) | 5 (1–17) |
| Cytology test | 1 (0–18) | 1 (1–18) | 0 (0–17) | 1 (1–11) | 3 (1–18) | 0 (0–9) | 4 (1–17) |
| HPV test | 2 (0–14) | 1 (0–11) | 2 (1–14) | 1 (0–10) | 2 (0–11) | 2 (1–8) | 4 (1–14) |
| Screening test | 2 (1–7) | 2 (1–7) | 2 (1–5) | 2 (1–5) | 2 (1–7) | 2 (1–5) | 2 (1–5) |
| Non-screening test | 0 (0–16) | 0 (0–16) | 0 (0–16) | 0 (0–9) | 1 (0–16) | 0 (0–10) | 3 (0–16) |
| Baseline result | | | | | | | |
| Positive | 548 (3.7%) | 144 (1.9%) | 404 (5.5%) | - | 144 (100%) | - | 404 (100%) |
| Screening history | | | | | | | |
| Previous abnormality | 1,208 (8.2%) | 584 (7.9%) | 624 (8.5%) | 566 (7.8%) | 18 (12.5%) | 570 (8.3%) | 54 (13.4%) |
| Previous CIN1+ | 925 (6.3%) | 444 (6.0%) | 481 (6.6%) | 428 (5.9%) | 16 (11.1%) | 439 (6.3%) | 42 (10.4%) |
| Previous CIN2+ | 404 (2.7%) | 187 (2.5%) | 217 (3.0%) | 180 (2.5%) | 7 (4.86%) | 197 (2.9%) | 20 (5.0%) |
| Without any screening history | 254 (1.7%) | 137 (1.9%) | 117 (1.6%) | 130 (1.8%) | 7 (4.9%) | 104 (1.5%) | 13 (3.2%) |
| ICC cumulative incidence proportion over 7 years (95% CI) | 0.08% (0.04%, 0.14%) | 0.10% (0.05%, 0.20%) | 0.06% (0.02%, 0.15%) | 0.10% (0.05%, 0.20%) | - | 0.01% (0.002%, 0.10%) | 0.76% (0.24%, 2.32%) |

Data are present $N$ (%), otherwise indicated.

All population: women participating at baseline; SD, standard deviation; Q1, first quartile; Q3, third quartile; CI, confidence interval; CIN1+, cervical intraepithelial neoplasia grade 1 or worse; CIN2+, cervical intraepithelial neoplasia grade 2 or worse; all lesions were diagnosed by histopathological test; previous abnormality, any previous positive test results before the trial, including HPV, cytology or histopathological test; non-screening test includes non-organized/opportunistic test or symptom-related indicated tests; ICC, invasive cervical cancer; HPV, human papillomavirus.

are all significantly higher than primary cytology test's sensitivity at year 5 (all $p < 0.0001$) (Table 3).

We calculated the IR of CIN2+ and ICC among the whole cohort, by arm, among women with primary negative results, and among women with or without previous abnormality

**Table 2. Follow-up participation by screening methods and baseline results N (%).**

| | Within 3 years | Within 6 years | Within 10 years |
|---|---|---|---|
| All population* | 2,394 (16.3%) | 11,018 (74.9%) | 12,936 (87.9%) |
| Primary cytology | 1,611 (21.8%) | 6,041 (81.6%) | 6,655 (89.9%) |
| Primary HPV | 783 (10.7%) | 4,977 (68.1%) | 6,281 (85.9%) |
| Cytology negative | 1,493 (20.6%) | 5,909 (81.4%) | 6,519 (89.8%) |
| Cytology positive | 118 (81.9%) | 132 (91.7%) | 136 (94.4%) |
| HPV negative | 495 (7.2%) | 4,595 (66.5%) | 5,895 (85.3%) |
| HPV positive | 288 (71.3%) | 382 (94.6%) | 386 (95.5%) |

*All population: women participating at baseline; HPV, human papillomavirus.

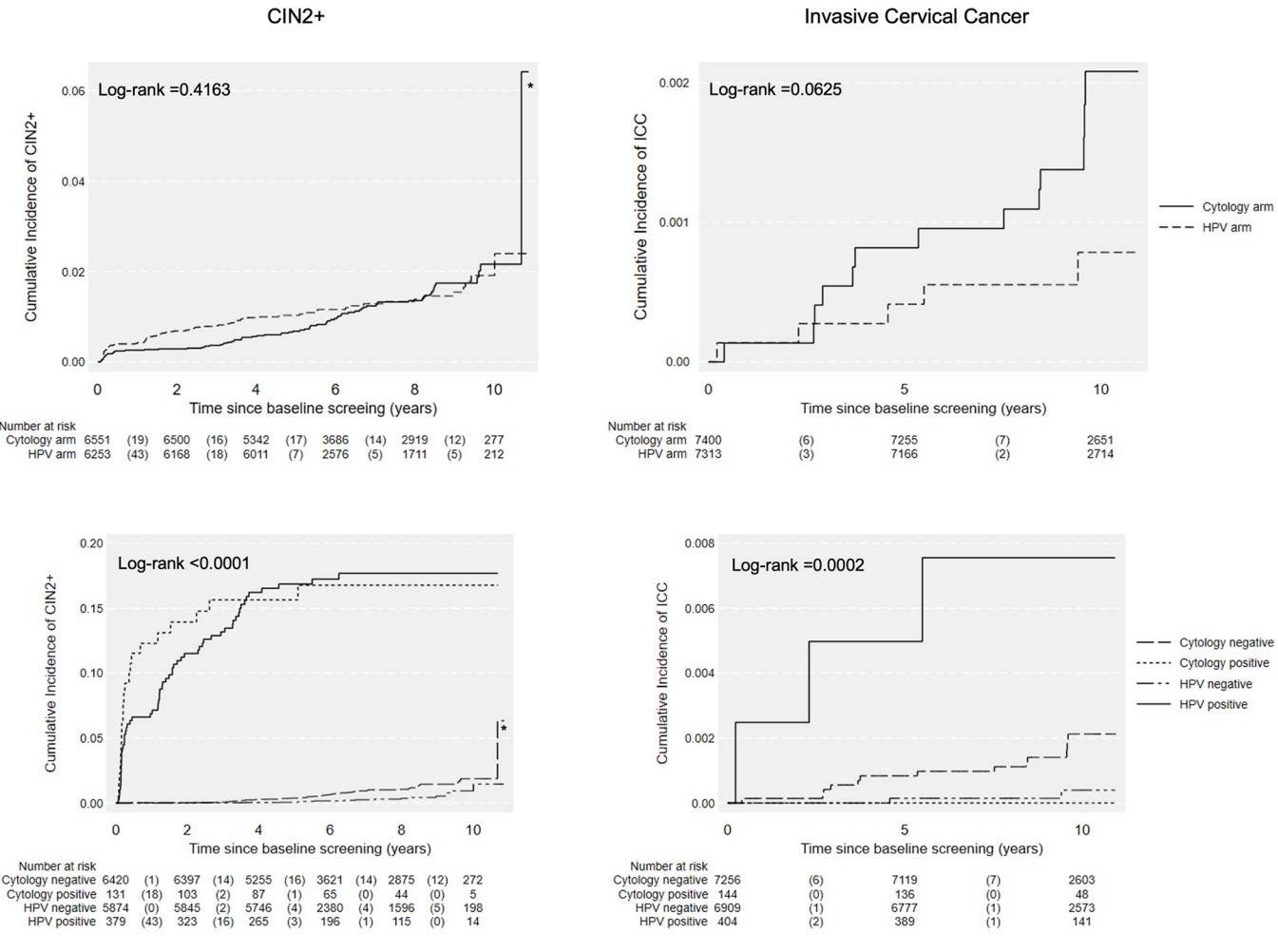

**Fig 2. Cumulative incidence of CIN2+ and ICC over 10 years by test methods and baseline results.** HPV, human papillomavirus; CIN2+, cervical intraepithelial neoplasia grade 2 or worse; ICC, invasive cervical cancer; follow-up time for CIN2+ was defined as the date of baseline test to the date of the last registered test. Death, emigration and total hysterectomy were considered as competing events for outcome ICC; * artificial increase due to censoring.

(Table 4 and Table B in S1 Text). Overall, the IRs (per 100,000 person-years) for CIN2+ were 189.6 (162.2 to 221.6), 180.4 (144.7 to 224.9), and 199.7 (160.2 to 249.0) for the whole cohort, in cytology arm, and in HPV arm, respectively. The IRs for ICC were 12.8 (8.1 to 20.3), 18.4 (10.7 to 31.7), and 7.2 (3.0 to 17.2) per 100,000 person-years for the whole cohort, in cytology arm, and in HPV arm, respectively.

Overall, point estimate for the risk of ICC over 10 years of follow-up among women aged 56 to 61 years in primary HPV arm was 0.39 compared to women in cytology arm in this trial, but this was not statistically significant (IRR: 0.39, 95% CI [0.14, 1.09], $p$ = 0.0726, Table 4) and had a similar risk of CIN2+ (IRR: 1.1, 95% CI [0.8, 1.5], $p$ = 0.5226, Table 4). Among women with baseline negative results, women with a primary HPV negative result had significantly lower risks of CIN2+ (IRR: 0.32, 95% CI [0.18, 0.55], $p$ < 0.0001, Table 4) and ICC (IRR: 0.16, 95% CI [0.04, 0.72], $p$ = 0.0163, Table 4). Among women without a previous abnormality, the risk of ICC was much lower among women with primary HPV test compared to women with primary cytology test (IRR: 0.23, 95% CI [0.07, 0.82], $p$ = 0.0237, Table B in S1 Text). The IR of

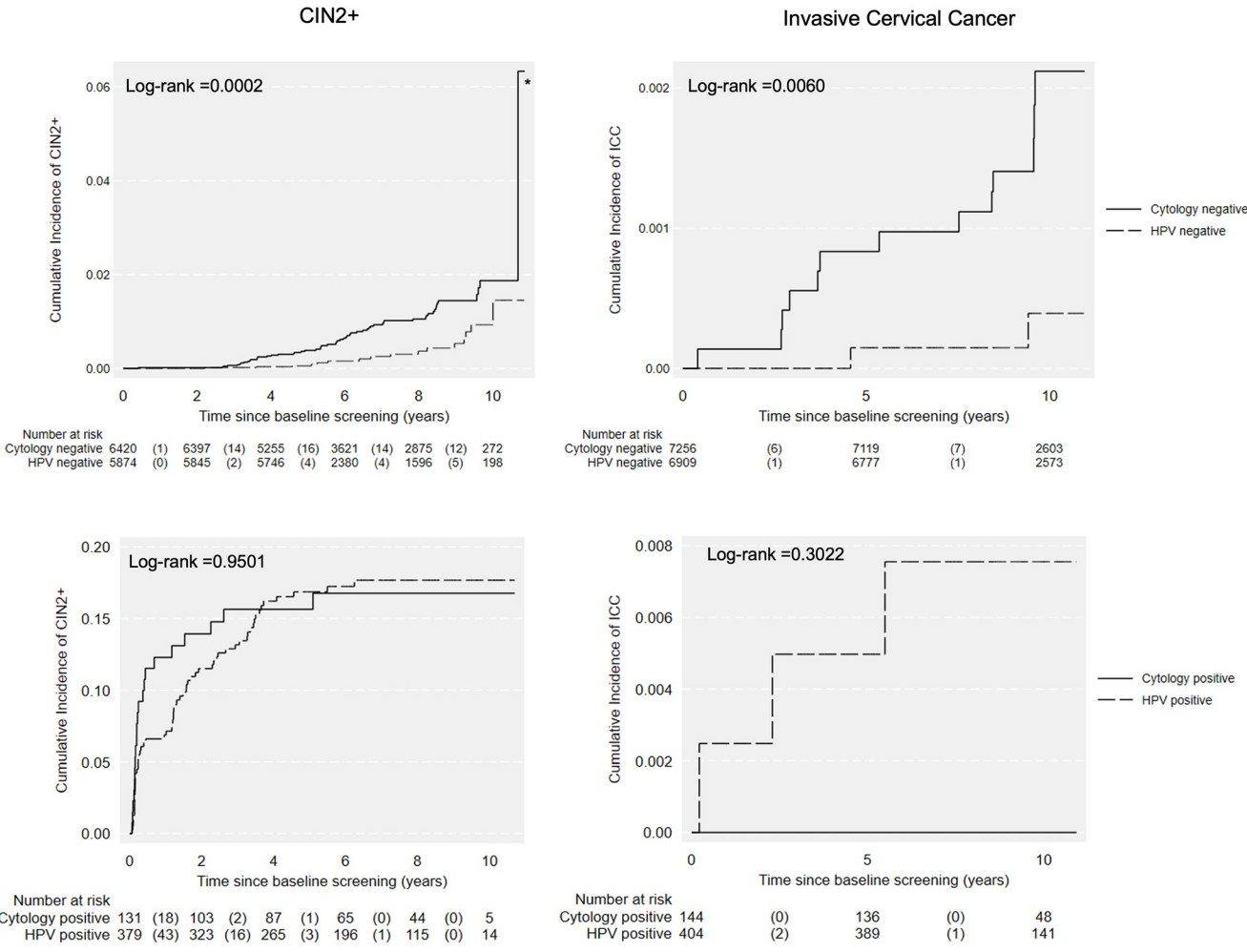

* Artificial increase due to censoring.

**Fig 3. Cumulative incidence of CIN2+ and ICC over 10 years by baseline results.** HPV, human papillomavirus; CIN2+, cervical intraepithelial neoplasia grade 2 or worse; ICC, invasive cervical cancer; follow-up time for CIN2+ was defined as the date of baseline test to the date of the last registered test. Death, emigration and total hysterectomy were considered as competing events for outcome ICC; * artificial increase due to censoring. The participation in testing during follow-up was 87.9% (12,936/14,713) for the whole cohort within 10 years. The participation in testing during follow-up was significantly higher in cytology arm compared to HPV arm, 89.9% (6,655/7,400) vs. 85.9% (6,281/7,313) ($p < 0.0001$) (Table 2).

ICC was 3.0 (0.8 to 12.1) among HPV arm baseline negative women and 18.8 (10.9 to 32.3) among cytology arm baseline negative women. We were unable to compare the IR among women with a previous abnormality between the arms because of the limited number of cases. The IR of ICC among women with positive baseline HPV result was 79.1 (25.5 to 245.3, Table 4). Adjusting IRR by sample year did not change the results (Table C in S1 Text).

We also compared the IRs between women with a previous abnormality and without a previous abnormality (Table D in S1 Text). Overall, women with a previous abnormality had higher risk of CIN2+ (IRR: 2.5, 95% CI [1.6, 3.9], $p < 0.0001$) and similar risk of ICC (IRR: 1.4, 95% CI [0.3, 6.1], $p = 0.6503$) compared to women without a previous abnormality. Among women in HPV arm, women with a previous abnormality had higher risks of CIN2+ (IRR: 2.2, 95% CI [1.1, 4.3], $p = 0.0198$) and ICC (IRR: 7.2, 95% CI [1.2, 43.1], $p = 0.0305$) compared to women without a previous abnormality. Among women with primary HPV positive result,

**Table 3. Longitudinal test characteristics using cervical intraepithelial neoplasia grade 2 or worse (CIN2+) as outcome at 3, 5, 7, and 10 years of follow-up.**

| | Cumulative counts of CIN2+ at follow-up by baseline results | | | | Test characteristics for CIN2+ | | | |
| --- | --- | --- | --- | --- | --- | --- | --- | --- |
| | Cytology | | HPV | | Cytology | | HPV | |
| | Negative | Positive | Negative | Positive | Sensitivity (95% CI) | Specificity (95% CI) | Sensitivity (95% CI) | Specificity (95% CI) |
| Baseline | 7,122 | 139 | 6,766 | 392 | | | | |
| 3 years | 4 | 20 | 1 | 49 | 84.5 (75.6, 90.5) | 98.2 (98.1, 98.2) | 98.1 (87.8, 99.7) | 94.3 (94.3, 94.4) |
| 5 years | 21 | 20 | 3 | 61 | 50.9 (46.7, 55.1) | 97.8 (97.8, 97.9) | 95.7 * (91.9, 97.8) | 92.2 (92.2, 92.3) |
| 7 years | 41 | 21 | 8 | 63 | 35.9 (33.0, 38.8) | 98.1 (98.1, 98.2) | 89.6 * (86.4, 92.1) | 91.2 (91.1, 91.2) |
| 10 years | 57 | 21 | 15 | 63 | 28.6 (25.0, 32.5) | 97.9 (97.7, 98.0) | 82.1 * (78.5, 85.3) | 92.2 (91.9, 92.4) |

*$P$-values for test of difference of proportions, compared to sensitivity of cytology at 5 years all $P < 0.0001$.

HPV, human papillomavirus; CI, confidence interval.

women with a previous abnormality had a similar risk of CIN2+ (IRR: 1.8, 95% CI [0.9, 3.5], $p = 0.0916$) and higher risk of ICC (IRR: 13.5, 95% CI [1.2, 148.4], $p = 0.0338$). The IRRs of CIN2+ of women with a previous abnormality compared to women without a previous abnormality in cytology arm were similar to women in HPV arm. Due to limited number of cases, we were unable to compare the IRs of ICC in cytology arm.

We calculated the IRs of ICC of women with or without organized screening test after receiving a baseline negative test result (Table E in S1 Text). The IRs for ICC in cytology arm were 13.4 (6.7 to 26.7) and 52.6 (21.9 to 126.5) among women with or without organized screening test. The IRs for ICC in HPV arm were 1.9 (0.3 to 13.5) and 7.5 (1.1 to 53.2) among women with or without organized screening test.

**Table 4. Incidence rate, incidence rate ratio, and 95% CI of cervical intraepithelial neoplasia grade 2 or worse (CIN2+) and invasive cervical cancer (ICC).**

| | CIN2+ | | | | | Invasive cervical cancer | | | | |
| --- | --- | --- | --- | --- | --- | --- | --- | --- | --- | --- |
| | $n$ * | Number of cases | IR (/100,000 person-years) | IRR | $P$-value | $n$ | Number of cases | IR (/100,000 person-years) | IRR | $P$-value |
| All population | 14,419 | 158 | 189.6 (162.2, 221.6) | | | 14,713 | 18 | 12.8 (8.1, 20.3) | | |
| Cytology | 7,261 | 79 | 180.4 (144.7, 224.9) | Ref. | | 7,400 | 13 | 18.4 (10.7, 31.7) | Ref. | |
| HPV | 7,158 | 79 | 199.7 (160.2, 249.0) | 1.1 (0.81, 1.5) | 0.5226 | 7,313 | 5 | 7.2 (3.0, 17.2) | 0.39 (0.14, 1.09) | 0.0726 |
| Women with negative baseline result | | | | | | | | | | |
| Cytology | 7,122 | 58 | 134.7 (104.1, 174.2) | Ref. | | 7,256 | 13 | 18.8 (10.9, 32.3) | Ref. | |
| HPV | 6,766 | 16 | 42.8 (26.2, 69.9) | **0.32 (0.18, 0.55)** | <0.0001 | 6,909 | 2 | 3.0 (0.8, 12.1) | **0.16 (0.04, 0.72)** | **0.0163** |
| Women with positive baseline result | | | | | | | | | | |
| Cytology | 139 | 21 | 2,886.3 (1,881.9, 4,426.8) | Ref. | | 144 | 0 | 0 | Ref. | |
| HPV | 392 | 63 | 2,844.9 (2,222.4, 3,641.8) | 1.0 (0.60, 1.6) | 0.9543 | 404 | 3 | 79.1 (25.5, 245.3) | ∞ | - |

*$n$ for CIN2+ exclude women with histopathological diagnosed CIN2+ before the randomized trial; HPV, human papillomavirus; CI, confidence interval; IR, incidence rate; IRR, incidence rate ratio; ref, reference; all population, women participating at baseline.

## Discussion

We performed a long-term evaluation of women ages 56 to 61 years who received a primary HPV-based screening in 2012 to 2014 and followed their risk of invasive cervical cancer over 10 years of follow-up, compared to women who received primary cytology-based screening within the same trial. Among women with a negative test result at baseline, women in the primary HPV arm had an 84% lower risk of ICC compared to women in the primary cytology arm. Among women who tested negative for HPV, the long-term ICC incidence rate (3.0/100,000 person-years) was below the elimination threshold (4/100,000 person-years) [3]. In contrast, it was 18.8/100,000 person-years for cytology negative women, which indicates it is not safe to exit older women with only a cytology negative test from screening. In addition, HPV testing had a substantially higher sensitivity of detecting CIN2+ within a 7-year interval compared to cytology test within a 5-year interval.

The overall long-term relative risk of ICC comparing primary HPV-based screening to primary cytology-based screening did not reach statistical significance in our current study, most likely due to a substantial part of the cytology arm subsequently being re-tested with the more sensitive method of primary HPV test as a result of a change in national recommendations for screening among older women in 2017—which affected all women equally in the nation. In a separate follow-up study of a similar guideline change in all women above 30 years, we could show that primary HPV-based screening overall is formally superior to primary cytology in cervical screening [25].

A few other studies have investigated the incidence rate of ICC among older women. Vahteristo and colleagues reported in a Finnish randomized trial that the incidence rate of ICC in the 2 groups—women screened with HPV and women screened with cytology was comparable over 15 years in women ages 25 to 65 years and among women ages 55 to 65 years the IR of ICC was 7.9/100,000 person-years [13]. The overall IRs of ICC were comparable between our cohort and the Finnish cohort in the same age group. In our study, however, women with primary HPV negative result had a statistically significantly lower risk of ICC compared to women with primary cytology negative result. This difference may be explained by the larger size of our trial in this age group and by factors such as our program using a different HPV-testing platform, the more sensitive cobas 4800, as compared to the Hybrid Capture 2 method used in the Finnish study [26–28]. Schroll and colleagues reported comparable IRs of ICC between HPV negative and cytology negative groups in Danish women ages 60 to 64 years, approximately 4/100,000 person-years [17]. However, the follow-up duration of this study was only 4 years, whereas we were able to follow women up to 10 years.

To our knowledge, we are the first study to report the incidence rate of ICC in this age group after a primary HPV positive and cytology positive test. Women with a cytology positive baseline result had 0 cases of ICC and women with HPV positive baseline result actually were diagnosed with 3 cases. This suggests the clinical management after cytology positivity was adequate and succeeded in preventing early lesions from progression. However, due to the limited number of cytology positive women ($n = 144$), more evidence is needed. All women with a positive cytology were immediately referred to colposcopy and histopathology test. The higher cancer incidence in the HPV arm may be caused by the less optimal triage strategy in this arm. Women with positive HPV results in 2012 were subjected to a reflex triage cytology test regardless of the HPV genotype [19]. If they were cytology negative, they were not necessarily referred to colposcopy and histopathology test. However, it is now known that women with HPV16/18 positive have higher risk of pre-cancer and cancer compared to women with other high-risk HPV positives [29]. Recent guidelines have already recommended direct referral to colposcopy for women tested positive for HPV 16/18 even when the cytology is negative

[30]. We thus posit that improved management of the HPV-positive women back in 2012 could likely have averted those cases.

To our knowledge, this is the first study to compare the longitudinal characteristics of HPV and cytology in detecting CIN2+ among older women by different time intervals. Elfström and colleagues reported the longitudinal characteristics of HPV and cytology in women aged 32 to 38 years and found the sensitivity of HPV to detect CIN2+ in 5 years was comparable to the sensitivity of cytology in 3 years [31]. We further found that the sensitivity of HPV detecting CIN2+ was much higher than cytology among women ages 56 to 61 years. In addition, as the sensitivity of cytology decreased from 85.9% in 3 years to 68.0% in 5 years among women aged 32 to 38 years in Elfström's study, the sensitivity of cytology dropped more drastically in our population [31]. This finding was in accordance with previous studies that have found a lower sensitivity for cytology among older women [5]. This may be caused by sampling error by a retracted transformation zone and interpretation problems due to atrophy of the cervical epithelium among older women [4]. We posit that the high sensitivity of primary HPV testing over a long time interval shows that high test positivity in the primary HPV arm was not merely overdiagnosis, but rather early detection of significant disease. Our data support that primary HPV-based screening is more reliable in protecting against ICC than primary cytology-based screening and older women may benefit more from changing the primary test from cytology to HPV.

Our results support the current Swedish guideline for cervical screening, which recommends HPV-based screening with a 7-year interval for women aged over 50 years up until 64 to 70 years. We found that women who tested cytology negative at the age of 56 to 61 years still had a high IR of ICC over 10 years of follow-up, which indicates the upper limit age in any primary cytology-based screening needs to be increased to over 60 years. The increase of the upper limit age in HPV-based screening needs more consideration. Among women who tested baseline negative for HPV, we found they had an IR of ICC lower than the WHO's elimination threshold, especially among women with another screening test after the trial baseline test. Women with one more organized screening test after baseline had a higher chance of being diagnosed with a precancer lesion and receive subsequent treatment. In contrast, among women without further screening after the trial baseline test, the IR of ICC remained high. In the era of transition from primary cytology-based screening to primary HPV-based screening, we think it would be safer to extend the upper limit age to over 60 years old. For women who start HPV-based screening at younger age (e.g., 30 years old), more evidence is needed to set up the upper limit screening age.

The low IR of ICC over 10-year and the high longitudinal sensitivity of primary HPV testing support the extension of screening interval to 7 years, even to 10 years. However, evidence from both our study and the POBASCAM trial indicates that women with previous abnormal screening history have a higher risk of precancer lesion during long-term follow-up [32]. Risk stratification based on previous screening history may be motivated. Due to the limited number of ICC cases in our study, we were unable at this point to provide more stratified results on the risk of ICC among women with a previous abnormality.

This randomized healthcare policy trial was originally designed and analyzed as a non-inferiority trial to evaluate the effectiveness of HPV testing in detecting CIN2+ within 24 months [14]. As the detection of the CIN2+ in the baseline would affect the risk CIN2+ and ICC during the follow-up, the non-inferiority hypotheses is no longer applicable for our follow-up period. We therefore focused on long-term effectiveness of primary HPV-based screening against ICC as a function of original trial arm. This study is the first long-term follow-up of primary HPV-based screening using ICC as the main outcome among older women. As we followed a randomized trial, there was little risk of selection bias. Data were retrieved from a

complete high-quality register of cervical tests, which minimized the risk of information bias. However, there are limitations. First, we detected a partial imbalance of invitations during follow-up between the original primary HPV and primary cytology arm, which caused women from the original primary cytology arm to have a shorter time interval between follow-up screening tests, and somewhat higher participation during the follow-up. This imbalance existed among sample taken in 2012 and 2013 (Table A in S1 Text), therefore we adjusted for sample year in our analyses. When these women were re-invited for repeat screening, the program had switched to primary HPV-based screening only in this age group. As all the women re-invited and re-tested during follow-up at that point received the more sensitive new method, the primary HPV test, the high participation, and shorter time interval could actually to some degrees have provided higher protection against ICC among women who were initially randomized to cytology arm. Thus, our analyses have likely somewhat underestimated the actual effectiveness of primary HPV-based screening against ICC among older women. Second, due to the limited number of women with a previous abnormality, we could not stratify the previous abnormality based on timescale since said lesion, which may have led to non-differential misclassification and also slightly underestimated the effectiveness of primary HPV-based screening. Third, despite the long follow-up, we only identified 18 ICC cases in total, which led to less precision in our analyses. However, the risk of ICC was still statistically significantly lower in the primary baseline HPV negative group. Finally, because of the small number of ICC cases in our study, for reasons of data privacy we could not present details on stage and whether the cancer was screening- or symptom detected. This should remain the focus of future work.

Our results show a primary negative HPV result can provide much higher reassurance against invasive cervical cancer than a primary negative cytology result and support the extension of the screening interval with HPV test in this age group. We further show that older women with primary negative HPV result had an incidence of ICC below the elimination threshold specified by WHO [3], especially among women without a previous abnormality. However, women with primary cytology negative result have an incidence of ICC substantially higher than the elimination threshold after 10 years. Furthermore, among women with a previous abnormality the incidence rates of CIN2+ and invasive cervical cancer were generally higher; among women with the last organized screening test around 56 to 61 years the incidence rate of invasive cervical cancer was still higher than 4/100,000. This suggests using primary HPV-based screening, risk stratification based on previous screening history and considering screen beyond 61 years old when releasing women from screening program. More detailed evidence on exit strategy would be highly utile to ensure the safety of older women, especially for those with adverse screening history.

In summary, our study supports that primary HPV-based screening is more reliable than primary cytology-based screening for detection of CIN2+ and prevention of ICC among older women. Our results also support the change of the screening strategy, for women over 50 years old, from primary cytology test with 5 years interval to primary HPV test with 7 years interval. A negative cytology test alone does not provide adequate reassurance against cervical cancer to cease screening among older women.

## Supporting information

**S1 CONSORT Checklist. CONSORT, Consolidated Standards of Reporting Trials.**
(DOC)

**S1 Protocol. Randomized Implementation of Primary HPV Testing in the Organized Screening for Cervical Cancer in Stockholm.** HPV, human papillomavirus.
(PDF)

**S1 Fig. Process flowchart of cervical screening for women in this trial.** CIN2+, cervical intraepithelial neoplasia grade 2 or worse.
(TIFF)

**S1 Text. Supplementary protocol, statistical analysis and tables.**
(DOCX)

**S1 Statistical Analysis Plan. Effectiveness of primary HPV-based cervical screening among older women: Long-term follow-up of a randomized healthcare policy trial.**
(PDF)

## Author Contributions

**Conceptualization:** Qingyun Yao, Jiangrong Wang, Karin Sundström.

**Data curation:** Qingyun Yao.

**Formal analysis:** Qingyun Yao, Jiangrong Wang.

**Funding acquisition:** Qingyun Yao, Karin Sundström.

**Investigation:** Qingyun Yao, Jiangrong Wang, K. Miriam Elfström, Björn Strander, Joakim Dillner, Karin Sundström.

**Methodology:** Qingyun Yao, Jiangrong Wang, K. Miriam Elfström, Joakim Dillner, Karin Sundström.

**Resources:** K. Miriam Elfström, Joakim Dillner, Karin Sundström.

**Supervision:** Jiangrong Wang, K. Miriam Elfström, Björn Strander, Joakim Dillner, Karin Sundström.

**Validation:** Qingyun Yao, Jiangrong Wang, K. Miriam Elfström, Joakim Dillner, Karin Sundström.

**Visualization:** Qingyun Yao.

**Writing – original draft:** Qingyun Yao.

**Writing – review & editing:** Qingyun Yao, Jiangrong Wang, K. Miriam Elfström, Björn Strander, Joakim Dillner, Karin Sundström.

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
