## [Editor Report · Decision Letter 0]

19 Jun 2024

Dear Dr Yao, 

Thank you for submitting your manuscript entitled "Effectiveness of primary HPV-based cervical screening among older women: Long-term follow-up of a randomised healthcare policy trial on exit testing" for consideration by PLOS Medicine.

Your manuscript has now been evaluated by the PLOS Medicine editorial staff and I am writing to let you know that we would like to send your submission out for external peer review.

Please re-submit your manuscript within two working days, i.e. by Jun 21 2024.

Feel free to email me at atosun@plos.org or us at plosmedicine@plos.org if you have any queries relating to your submission.

Kind regards,

Alexandra Tosun, PhD

Associate Editor

PLOS Medicine

---

## [Decision Letter · Decision Letter 1]

29 Jul 2024

Dear Dr Yao,

Many thanks for submitting your manuscript "Effectiveness of primary HPV-based cervical screening among older women: Long-term follow-up of a randomised healthcare policy trial on exit testing" (PMEDICINE-D-24-01926R1) to PLOS Medicine. The paper has been reviewed by subject experts and a statistician; their comments are included below and can also be accessed here: [LINK]

As you will see, the reviewers were positive about the paper, but raised a number of questions about specific details of the study, including the methodology and, for example, how the results might translate into screening guidelines. After discussing the paper with the editorial team, I'm pleased to invite you to revise the paper in response to the reviewers' comments. We plan to send the revised paper to some or all of the original reviewers, and we cannot provide any guarantees at this stage regarding publication.

We ask that you submit your revision by Aug 19 2024. However, if this deadline is not feasible, please contact me by email, and we can discuss a suitable alternative.

Don't hesitate to contact me directly with any questions (atosun@plos.org). 

Best regards, 

Alex

Alexandra Tosun, PhD 

Associate Editor

PLOS Medicine

atosun@plos.org

Comments from the reviewers: 

Reviewer #1: I have not identified any issues that need to be addressed.

Reviewer #2: Thanks for the opportunity to review the paper entitled "Effectiveness of primary HPV-based cervical screening among older women: Long-term follow-up of a randomized healthcare policy trial on exit testing"

The paper describes risk of CIN2+ and cervical cancer in older women who underwent cytology vs HPV-based screening as part of a clinical trial. The authors conclude that an HPV-negative test provides greater re-assurance against cervical cancer than a negative cytology.

I find that the paper is important and interesting. Results are clinically relevant. Overall, the paper is well written but there are typos and missing words occasionally (P 6, l 112; P25, l 326; p26, line 363, etc), and some phrases are hard to understand. Please find below my detailed comments, which I hope will be useful for the authors.

1. Title: Consider deleting "on exit testing".

2. Abstract: 

a. second sentence in the background is long and hard to understand. Please revise/split up. 

b. Result headline is missing. 

c. Please revise the following. Either delete an 84% or delete statistically significant : "Among women with a negative test result at baseline, women in THE primary HPV arm had an 84% statistically significant lower risk of ICC compared to women in the primary cytology arm (IRR: 0.16; 95% CI: 0.04, 0.72). " 

3. Introduction:

a. P5, lines 75-76: another (and very important) reason is the diagnostic difficulties in this group of women. This is often an overlooked issue and should be stated here.

b. P5, line 81: I guess it differs between countries and not women?

c. P6, line 96. Delete previously (redundant)

d. P6, line 101. This sentence is unclear to me. Please elaborate. This seems to be important information to the reader, and readers should not have to read another paper to understand. 

4. Methods

a. P7, lines 115-117. This is unclear. Please specify.

b. P7, line 124: CIN1 is a histologic diagnosis and should not be used for cytological grading. Please delete.

c. P8, line 140: CIN2+?? How can women have a histological diagnosis of CIN2+. Do you mean high-grade cytology? (HSIL, AIS, AGC, ASC-H)?

d. P8, lines 142: Please elaborate. As only women who have a biopsy collected are at risk of CIN2+ and ICC, it is important to know if all women have one or multiple biopsies collected or if biopsies are collected at the discretion of the colposcopist. Also, it is important to know if LLETZ were part of the diagnostic work-up as well or if you only collected biopsy results. A previous study on women in the same age group has revealed that biopsies (when collected) miss 54% of CIN2+ cases.

e. P8, line 159. Please continue to use the Bethesda classification. Thus, replace koilocytosis with ASCUS+

f. P8, line 161-p9, 169. This section is confusion and needs to be rephrased. I assume you are referring to the two analyses; CIN2+ and ICC. Follow-up did not start after their diagnosis, right? But rather women contributed time at-risk from baseline until a record of CIN2+ and ICC, respectively. You only state censoring events for ICC, but I assume the same censoring events are applicable to the CIN2+ analysis (death, emigration, hysterectomy). Also, women who undergo hysterectomy due to benign disease should be censored and not just the ones that had a hysterectomy due to cancer. Also, please specific what CIN2+ includes (I assume, CIN2, CIN3, AIS, and all cancers). 

g. It would be helpful for the reader if you could state the primary and secondary outcome 

h. I would be helpful to know more about previous screening history. How was this information collected and defined? Any abnormal cytology or histology only??

i. Statistics is very long. Suggest condensing.

j. Why exclude women with a previous history of CIN2+

5. Results

a. It would be helpful if you could start with a description of how many were enrolled and any differences in important baseline characteristics across arms. The word median is used in almost every sentence in the beginning of the results. Please revise.

b. I am worried about the use of sensitivity and specificity at 3, 5, 7, 10 years given that these estimates are all based on the baseline test result. In the clinical world, a clinician needs to know how sensitive and specific a test is to detect disease at the time of the test and not after 10 years. I think it is misleading. A lot can happen over the course of 10 years.

c. P 14, line 236-241: During follow-up?

6. Discussion:

a. It would be nice if the first sentence had been stated earlier in the manuscript.

b. I think it is important to discuss the risk of verification bias across groups. Only women who are referred for colposcopy and have biopsies collected are at risk.

c. P. 28, lines 412-414. Please revise to improve readability.

d. I think it would be interesting to discuss a bit more why you did not find any cancers in those with a positive cytology. Could it be because they were more likely to get treatment, thereby avoiding progression?

7. Figure 1. Please specify what "pathology" means. Cervical biopsy and LLETZ? Hysterectomy?

8. Figure 2. Please state which groups are being compared in the log-rank test. When looking at the figures, I don't see any differences in HPV neg vs cytology neg women in terms of CIN2+, but there is a major difference between those that are pos vs those that are negative. In terms of cancer, it is evident that HPV-pos have a high risk, while cytology positive have lowest. The supplementary figure illustrates what the title legend says: there is a significant difference in HPV neg vs cyt neg. Please revise.

9. Supplementary table 2. Confusing to read. You need to include the two groups you are comparing wo it becomes clear and concise to the reader which one is the reference group. Same goes for Supplementary table 3. 

10. 

Reviewer #3: Primary HPV testing had a higher sensitivity of detecting CIN2+ within a 7-year interval than primary cytology test within a 5-year interval (89.6% vs 50.9%). 

This study also showed that older women with an HPV negative result had higher risk of subsequent CIN2+ if they had a previous cytology positive test result. These results indicate that leaving with a cytology negative test at the age of 60 in general is not safe, given the high incidence rate of ICC after 10 years, at least in women with previous abnormalities

1. Only 35% of countries worldwide 78 currently endorse HPV-based screening in their national recommendations. The recommended criterion of exiting program differs between women with only negative screening history within the past 10-year period, or women with one negative HPV test when arriving at the upper age limit. What is the case in most European countries? Can you please provide these data? 

2. There is also a lack of evidence of when it is appropriate to exit the screening. Most countries recommend women exit screening around 60-70 years old. This study supports the extending of age upper limit for cervical screening. Women with their last organized screening test above the age of 55 still have a risk of invasive cervical cancer higher than the elimination threshold. Till what age would the authors suggest screening? Would this be HPV-based, cytology based? Or a combination? 

3. This is one the first study that compared the longitudinal characteristics of HPV and cytology in detecting CIN2+ or ICC among older women by different time intervals. Table 3 shows longitudinal test characteristics using cervical intraepithelial neoplasia grade 2 or worse (CIN2+) as outcome at 3, 5, 7 and 10 years of followup. What would be the guideline interval? Based on what? Would this be different between women with previous abnormal smears? 

4. During the follow-up, women were invited women with a negative result from HPV arm or cytology arm to the next test after 5 years, women with HPV positive and cytology-negative result were invited to a repeat HPV and cytology test after a year (or after 3 years from May 2013); women with low-grade cytological abnormalities in primary cytology test and negative HPV triage were invited for a repeat test one year later. Women with persistent HPV infection (positive for the same HPV type category), with lowgrade cytological abnormalities and HPV positive triage, or with CIN2+, were referred to colposcopy. Could you please pout these data in a process flowchart? 

Reviewer #4: This is a well conducted study on the effectiveness of primary HPV-based cervical screening among older women from the long-term follow-up of a randomised healthcare policy trial on exit testing. The study design, statistical methods and analyses, and presentation and interpretation of the results are mostly adequate. However, there are still a few issues needing attention.

1. Throughout the paper, it hinted the HPV screening reduced the risk of cervical cancer, for example, in abstract, it says "Overall, women in the primary HPV arm had a tendency of a lower risk of ICC compared to women in cytology arm (IRR: 0.39; 95% CI: 0.14, 1.09)". However, the HPV screening is just diagnostic test which will not reduce the risk itself. It needs to be clear, it is the whole package - early and effective detection, early intervention, which may lead to reduction in the risk and improvement in the outcomes in the long term. It would be adquate to use 'associated' instead, and clearly discuss what exactly contribute the lower risk of ICC after the screening in the long run.

2. Competing risk. As the outcome in the survival analyses is CIN2+ or ICC other than all cause mortality, there might be an issue of competing risk from death in the analyses. Need to know total number and percentage of death in the cohort during the follow up.

3. Figure 2 is a key figure. For the 4 groups comparison results, the log rank test gives an overall significane of the comparison. However, we are also interested in those close curves, for example, in figure 2c, HPV postitive vs cytology positive, HPV negative vs cytology negative. Can authors please provide these mentioned test results too? 

Reviewer #5: This is an important trial report on a topic for which there has been limited evidence to date - optimising cervical screening in older women. The setting for this longer-term observational follow-up of a trial is Sweden, where there is excellent capacity for case ascertainment and large-scale linkage capability. 

Women aged 56-61 were randomised in 2012-14, with samples randomised in the laboratory to cytology with HPV triage or HPV screening with cytological triage. HPV testing with HPV16/18 partial genotyping was used. Women with a negative result were invited to attend again in 5 years. A total of 14,719 women participated. 

Overall, the IRR for CIN2+ was similar for the two arms, but much lower in women with a negative baseline result of HPV. For ICC the IRR was significantly lower in HPV negative vs cytology negative women.

The longitudinal sensitivity of cytology clearly dropped over the follow-up time, whereas this was not the case to the same extent for HPV screening.

MAJOR COMMENTS

* It is surprising that a relatively small proportion of women at this age had a previous abnormality. Could more detail be provided about the look back period. Was lifetime screening behaviour or just very recent behaviour available (or something in between)? Were women who were previously unscreened considered differently from those who were well screened but without an abnormality? It is suggested the authors consider these issues for sensitivity analysis. 

* The moderate imbalance of follow-up between the two study arms is potentially an issue because in this observational follow-up design there is no consistent ascertainment of whether disease is present at the end of follow-up. Therefore, ascertainment bias for the outcome of invasive cervical cancer (as well as CIN2+ and CIN3+) is a possibility especially in respect to early stage non-symptomatic cancers. It is somewhat reassuring that biopsy rates appeared fairly similar. However, the IRs of ICC tended towards being different in women who received an organised test after the baseline test than those that did not. Could this latter issue be discussed and explained in more detail? 

* Can explicit detail be provided on the 18 cancers on the mode of detection - especially Stage/sub-stage at diagnosis and whether symptomatic or screen-detected. A table could ideally be provided on this showing classifications for each cancer. It is important to know which cancers are screen 'successes' i.e. screen/triage/fu detected, or screen 'failures' i.e. interval cancers detected symptomatically.

* No detail on cancer stage is provided but microinvasive cancers would normally be only screen detected and it would be good to see all the results also presented with these excluded and with a frank cancer outcome only. Is staging at diagnosis data available? If not, this reason for this data limitation should be explained and the implications discussed as a limitation of the analysis. 

* Particularly, given that the IR of ICC in women with a positive HPV test was apparently quite high (albeit with only three cases and thus high uncertainty in the estimate) it is important to provide more detail on the three cancers in this group? If there was evidence that these cases represented early stage microinvasive cancers rather than frank cancers and they were screen-detected, then the inference could be made that these were detected earlier because of screening.

* The elimination threshold is not really designed to be used in the way it is here - i.e. as a threshold for the implied safety of discharge of women. Elimination is a population-wide average rate and population-wide target. Although it is an interesting to compare outcomes in groups here to elimination rates, I would suggest not relying on this for the primary conclusions. The acceptability of screening discharge in a population subgroup post-testing does depend on the rate of disease thereafter but the post-test probability should be compared to the disease rates for women routinely referred to colposcopy, follow-up, or routine screening in that setting. Different countries have different acceptable disease rates for these thresholds.

MINOR COMMENTS

* The second peak referred to in the first sentence of the introduction is most likely because of screening stopping and hence varies by screening stopping age in a population - this should be made clearer.

* In Table 1, are the non-screening tests including both triage and follow-up tests? Can this be made clearer or broken down?

* Please upload any figures associated with your paper as individual TIF or EPS files with 300dpi resolution at resubmission; please read our figure guidelines for more information on our requirements: http://journals.plos.org/plosmedicine/s/figures. While revising your submission, please upload your figure files to the PACE digital diagnostic tool, https://pacev2.apexcovantage.com/. PACE helps ensure that figures meet PLOS requirements. To use PACE, you must first register as a user. Then, login and navigate to the UPLOAD tab, where you will find detailed instructions on how to use the tool. If you encounter any issues or have any questions when using PACE, please email us at PLOSMedicine@plos.org.

* DATA AVAILABILITY: The Data Availability Statement (DAS) requires revision. Please clarify whether the data are freely/publicly available, owned by a third party but freely available upon request, or not freely available (if so, please briefly describe the ethical, legal, or contractual restriction that prevents you from sharing the data).

* COMPETING INTEREST: All authors must declare their relevant competing interests per the PLOS policy, which can be seen here: https://journals.plos.org/plosmedicine/s/competing-interests

For authors with ties to industry, please indicate whether any of the interests has a financial stake in the results of the current study.

* Please include the completed CONSORT checklist as Supporting Information. When completing the checklist, please use section and paragraph numbers, rather than page numbers.

FIGURES AND TABLES

SUPPLEMENTARY MATERIAL

REFERENCES

STUDY TYPE-SPECIFIC REQUESTS

* Please report your study according to CONSORT, but explicitly state that it is a sub-study. Please ensure that the Abstract describes the main points of the original trial (2-3 sentences, including study population, study dates, intervention and primary outcome), but that the main part of the Abstract describes the details of the follow-up study.

* Please ensure that study registration details are included in the Methods section.

* Abstract: Please include the study design, population and setting, number of participants, years during which the study took place (original enrollment and follow up), length of follow up, and main outcome measures.

* Please include absolute numbers wherever you report percentages; eg, n/N (%)

* In keeping with our commitment to Open Science, please include the study protocol document and analysis plan (including any amendments) as Supporting Information to be published with the manuscript if accepted.

---

## [Decision Letter · Decision Letter 2]

18 Nov 2024

Dear Dr. Yao,

Thank you very much for re-submitting your manuscript "Effectiveness of primary HPV-based cervical screening among older women: Long-term follow-up of a randomised healthcare policy trial" (PMEDICINE-D-24-01926R2) for review by PLOS Medicine.

Firstly, I would like to apologise for the fact that the review process has taken longer than usual. We would like to thank you for your patience.

Thank you for your detailed response to the editors' and reviewers' comments. I have discussed the paper with my colleagues and the academic editor, and it has also been seen again by two of the original reviewers. The changes made to the paper were satisfactory to the reviewers. As such, we intend to accept the paper for publication, pending your attention to the editors' comments below in a further revision. When submitting your revised paper, please once again include a detailed point-by-point response to the editorial comments.

[LINK]

In revising the manuscript for further consideration here, please ensure you address the specific points made by each reviewer and the editors. In your rebuttal letter you should indicate your response to the reviewers' and editors' comments and the changes you have made in the manuscript. Please submit a clean version of the paper as the main article file. A version with changes marked must also be uploaded as a marked up manuscript file. Please also check the guidelines for revised papers at http://journals.plos.org/plosmedicine/s/revising-your-manuscript for any that apply to your paper.

We ask that you submit your revision within 1 week (Nov 25 2024). However, if this deadline is not feasible, please contact me by email, and we can discuss a suitable alternative.

Please do not hesitate to contact me directly with any questions (atosun@plos.org). If you reply directly to this message, please be sure to 'Reply All' so your message comes directly to my inbox.

We look forward to receiving the revised manuscript.   

Sincerely,

Alexandra Tosun, PhD

Associate Editor 

PLOS Medicine

plosmedicine.org

Comments from Reviewers:

Reviewer #2: Thank you for your effort in revising your paper. You have adequately addressed all my concerns and comments. The paper reads well and is clinically very timely and relevant. Thanks!

Reviewer #4: Thanks authors for their effort to improve the manuscript. I am satisfied with the response and revision. No further issues needing attention.

[LINK]

Comments from Academic Editor:

The authors seem to have responded well to a number of questions and issues raised by the reviewers. The large size, initial randomisation and long follow-up make this paper to some extent unique, and the findings are important and have direct clinical implications. The statistically significant effects of screening modality on CIN2+ and ICC are meaningful, but so is the trend in the overall population, which did not reach statistical significance but has a confidence interval that shows strong evidence of an overall benefit.

Requests from Editors:

TITLE

We suggest changing the title to “Evaluation of primary HPV-based cervical screening among older women: Long-term follow-up of a randomised healthcare policy trial in Sweden”

CODE AVAILABILITY

Thank you for stating that code can be shared at request. Could you provide an appropriate contact (web or email address) for inquiries?

ABSTRACT

1) l.39: Please define ‘HPV’ at first use.

2) l. 51: Do you mean ‘primary HPV testing’?

3) l.57: Please refrain from describing non-significant results as showing “a tendency for”. Please revise throughout the manuscript.

4) l.58: Please define ‘CI’ at first use.

5) l.67, please change to: “For women over 55 years of age,…”. When presenting age, please add a unit, such as ‘years’. Please revise throughout the manuscript, including tables and figures (including those in the Supporting Information).

6) ll.67-69: Please rewrite the Abstract conclusion, as we feel that the phrase "substantially more reassuring" refers to the participants' perception, which was not assessed here. Please use more technical language; the phrase "In this study, we observed ..." may be useful.

7) In the last sentence of the Abstract Methods and Findings section, please describe the main limitation(s) of the study's methodology.

AUTHOR SUMMARY

1) Please revise the formatting; each sub-heading should contain 2-3 single sentence, concise bullet points.

2) In the final bullet point of 'What Do These Findings Mean?', please include the main limitations of the study in non-technical language. 

INTRODUCTION

1) l.95: Please define ‘IR’ at first use. 

2) l.106: Please define ‘ICC’ at first use.

METHODS AND RESULTS

1) l.190: Please define ‘AIS’ at first use.

2) Figure 1/2/3: Please define ‘HPV’ in the figure description.

3) Table 1: Please define ‘HPV’, ‘Q1-Q3’, and ‘ICC’ below the table.

4) Figure 3: Please note that in Figure 3 the explanation for the asterisk is missing.

5) Table 2/3/4: Please define ‘HPV’ below the table.

6) l.318: Please report statistical information as follows to improve clarity for the reader "22% (95% CI [13%,28%]; p</=). For example: “(IRR: 0.39, 95% CI [0.14,1.09], p=0.0726, Table 4)” When reporting 95% CIs please separate upper and lower bounds with commas instead of hyphens as the latter can be confused with reporting of negative values. Please revise throughout the manuscript.

7) ll.319-321, please change to: “Among women with baseline negative results, women with a primary HPV negative result had significantly lower IRs of CIN2+ (IRR: 0.32, 95% CI [0.18,0.55], p<0.0001) and ICC (IRR: 0.16, 95% CI [0.04,0.72], p=0.0163, Table 4).”

8) l.316ff: Please check carefully that you have used 'IR' or 'IRR' appropriately in the text. We have noticed that you have used the term 'IR' when presenting 'IRR' values in parentheses. Please revise throughout the entire main text. 

9) ll.332-339: Please revise according to comment 7 (i.e. two separate parentheses).

DISCUSSION

1) l.384: Please remove the word ‘respectively’.

2) l.415, please change to "HPV-based screening”.

3) l.486: Please remove the conclusion subheading. The Conclusion paragraph should be a continuous part of the Discussion section.

REFERENCES

For references 3 and 25, please remove one of the dates. We only need you to include the access date (e.g., [accessed: 10/04/2024]).

SUPPLEMENTARY MATERIAL

1) Please ensure to reference all supporting material in the main text.

2) Please note that in the S4 Support Information you have misspelled the heading "Screening Protocol".

3) In the published article, supporting information files are accessed only through a hyperlink attached to the captions. For this reason, you must list captions at the end of your manuscript file. You may include a caption within the supporting information file itself, as long as that caption is also provided in the manuscript file. Do not submit a separate caption file.

When SI files are contained with a single file:

Please label the file as ‘S1 Supporting Information’.

Please apply alphabetical labelling to each table and figure contained within the S1 file. For example, ‘Fig A’ to ‘Fig Z’ and ‘Table A’ to ‘Table Z’.

Plain text does not need to be labelled and can just be given a title as necessary. For example, ‘Statistical Analysis Plan’.

Please cite tables/figures as ‘Fig A in S1 Supporting Information’ and/or ‘Table A in S1 Supporting Information’, for example.

Please cite plain text as, ‘Statistical Analysis Plan in S1 Supporting Information’, for example.

When SI files are uploaded as separate files:

Please label tables as ‘S1 Table’ (so on) and figures as ‘S1 Fig’ (and so on).

Any additional documents (protocols/analysis plans etc.) can be labelled as ‘S1 Protocol’, for example. Please cite items as exactly as labelled.

SOCIAL MEDIA

To help us extend the reach of your research, please provide any X (formerly known as Twitter) handle(s) that would be appropriate to tag, including your own, your co-authors’, your institution, funder, or lab. Please enter in the submission form any handles you wish to be included when we post about this paper.

General Editorial Requests

---

## [Editor Report · Decision Letter 3]

27 Nov 2024

Dear Dr Yao, 

On behalf of my colleagues and the Academic Editor, Elvin Hsing Geng, I am pleased to inform you that we have agreed to publish your manuscript "Evaluation of primary HPV-based cervical screening among older women: Long-term follow-up of a randomised healthcare policy trial in Sweden" (PMEDICINE-D-24-01926R3) in PLOS Medicine.

I appreciate your thorough responses to reviewers' and editors' comments throughout the editorial process. Thank you again for your patience throughout the process. We look forward to publishing your manuscript, and editorially there are only a few remaining minor stylistic/presentation points that should be addressed prior to publication. We will carefully check whether the changes have been made. If you have any questions or concerns regarding these final requests, please feel free to contact me at atosun@plos.org.

Please see below the minor points that we request you respond to:

1) Author Summary: Please add bullet points/symbols.

2) l. 123, please define ‘HPV’ at first use.

3) Figure 1: Please note that in Figure 1 there is one instance where HPV is written in lower case letters (“Women with invalid hpv results”). Please revise.

4) l.514: Please note that the Conclusion subheading is still in the main text. Please remove.

5) Please note that there are still several instances of age being reported without the appropriate unit, i.e. years. Please check and revise carefully. A few examples:

l.140, please change to: “women aged 30-64 years”

l.141, please change to: “aged 56-61 years”

l.254, please change to: “until age 64 years”

l.261, please change to: “at age of 56-61 years”

ll.338-339, please change to: “women aged 56-61 years”

l.373, please change to: “women ages 56-61 years”

l.397, please change to: “in women ages 25-65 years and among women ages 55-65 years”

l.405, please change to: “Danish women ages 60-64 years”

Before your manuscript can be formally accepted you will need to complete some formatting changes, which you will receive in a follow up email (including the editorial points above). Please be aware that it may take several days for you to receive this email; during this time no action is required by you. Once you have received these formatting requests, please note that your manuscript will not be scheduled for publication until you have made the required changes.

PRESS

Sincerely, 

Alexandra Tosun, PhD 

Associate Editor 

PLOS Medicine